# Delayed correlation between the incidence rate of indigenous murine typhus in humans and the seropositive rate of *Rickettsia typhi* infection in small mammals in Taiwan from 2007–2019

Pai-Shan Chiang[1], Shin-Wei Su[2], Su-Lin Yang[1], Pei-Yun Shu[1], Wang-Ping Lee[2], Shu-Ying Li[1], Hwa-Jen Teng [1] *

1 Center for Diagnostics and Vaccine Development, Centers for Disease Control, Ministry of Health and Welfare, Taipei, Taiwan, 2 Division of Quarantine, Centers for Disease Control, Ministry of Health and Welfare, Taipei, Taiwan

* hjteng@cdc.gov.tw

## Abstract

Murine typhus is a flea-borne zoonotic disease with acute febrile illness caused by *Rickettsia typhi* and is distributed widely throughout the world, particularly in port cities and coastal regions. We observed that murine typhus was an endemic disease (number of annual indigenous cases = $29.23\pm8.76$) with a low incidence rate ($0.13\pm2.03*10^{-4}$ per 100,000 person-years) in Taiwan from 2007–2019. Most (45.79%, 174/380) indigenous infections were reported in May, June, and July. The incidence rates in both May and June were statistically higher than those in other months ($p<0.05$). Correspondingly, sera collected from small mammals (rodents and shrews) trapped in airports and harbors demonstrated anti-*R. typhi* antibody responses (seropositive rate = $8.24\pm0.33\%$). Interestingly, the ports with the highest seropositivity rates in small mammals are all inside/near the areas with the highest incidence rates of indigenous murine typhus. In addition, incidence rates in humans were positively correlated with the 1-month and 2-month prior seropositive rates in small mammals ($R = 0.31$ and 0.37, respectively). As early treatment with appropriate antibiotics for murine typhus could effectively shorten the duration of illness and reduce the risk of hospitalization and fatality, flea-related exposure experience should be considered in clinics during peak seasons and the months after a rise in seropositivity rates in small mammals. Surveillance in small mammals might be helpful for the development of real-time reporting or even early reminders for physicians of sporadic murine typhus cases based on the delayed correlation observed in this study.

**Data Availability Statement:** All relevant data are within the manuscript and its Supporting Information files.

**Funding:** The author(s) received no specific funding for this work.

**Competing interests:** The authors have declared that no competing interests exist.

## Author summary

Murine typhus is a flea-borne zoonotic disease with acute febrile illness caused by *Rickettsia typhi* and is distributed widely throughout the world, particularly in port cities and coastal regions. Early treatment with appropriate antibiotics for murine typhus could effectively shorten the duration of illness and reduce the risk of hospitalization and fatality. However, it presents with nonspecific symptoms and is oftentimes misdiagnosed. In Taiwan, murine typhus has been designated a notifiable disease since 2007. Meanwhile, surveillance of *R. typhi* infection of small mammals was also launched at 25 international airports and harbors. Since then, we observed that indigenous murine typhus patients have been detected in Taiwan annually and sera collected from small mammals trapped in ports also demonstrated anti-*R. typhi* antibody responses. Correspondingly, the ports with the highest seropositivity are all inside/near the areas with the highest incidence rate of indigenous murine typhus in Taiwan. We further found that incidence rates in humans were positively correlated with the 1-month and 2-month prior seropositive rates in small mammals. Surveillance in small mammals might be helpful for the development of real-time reporting or even early reminders of sporadic murine typhus cases based on the delayed correlation observed in this study.

## Introduction

Murine typhus (also known as endemic typhus fever) is a flea-borne zoonotic disease that is caused by a bacterium, *Rickettsia typhi*. This disease is distributed widely throughout the world, particularly in port cities and coastal regions [1]. The bacterium was discovered in Americans. The first suspected case was reported in 1913 in Atlanta [2]. Its flea-borne route of transmission was realized by a series of experiments, including for endemic typhus, which could be established in guinea pigs with brain emulsions of wild rats, and pulverized flea taken from wild rats in Mexico and the USA during the 1930s [3–6]. *R. typhi* has been detected in different animal hosts, such as rats, cats, and opossums [7,8]. Vectors including the Oriental rat flea (*Xenopsylla cheopis*) and cat flea (*Ctenocephalides felis*) could acquire the bacterium from the infected animal host throughout their life. This bacterium multiplies in the epithelial cells of the flea's midgut and is shed in fleas' feces. When a flea bites, the bite causes a wound on the skin of an animal or a person. At the same time, the flea also defecates, and its excrement with the pathogen can be rubbed into the bite wound or other wounds, causing infection [7,9,10]. In addition, *R. typhi* can remain infectious in dry feces for 100 days, and inhalation of contaminated aerosols can therefore also cause infections [11,12], but this pathogen will not spread from person to person. Moreover, the bacteria infect the flea's reproductive organs and could be passed to offspring by transovarial transmission [13,14].

In humans, symptoms of murine typhus range from short-term illness, including fever, headache, musculoskeletal pain, rash, and arthralgia, in mild cases to organ-specific complications (e.g., pneumonia, hepatitis, meningoencephalitis, myocarditis, endocarditis, splenic rupture, kidney dysfunction or a combination) and multiple organ failure [8,11,12,15–22]. These clinical features are nonspecific and diverse and are similar to those of several common febrile illnesses, including serious non-vector-associated infections (e.g., typhoid fever, invasive meningococcal disease, leptospirosis, viral and bacterial meningitis, measles, toxic shock syndrome, and secondary syphilis) and vector-borne infections (e.g., Dengue, West Nile fever, Rocky Mountain spotted fever, ehrlichiosis, anaplasmosis, and babesiosis) [13,23,24]. Therefore, murine typhus is an easily misdiagnosed disease. However, delays in murine typhus

treatment can extend the duration of symptoms and increase the risk for complications (e.g., seizures, respiratory failure, and persistent frontal and temporal lobe dysfunction) or death [8,16]. The mortality rate for murine typhus is 4% and could be reduced to 1% with the use of appropriate antibiotics [16]. Hence, early treatment with appropriate antibiotics against murine typhus plays a pivotal role in prognosis. Prompt treatment for suspected cases without laboratory confirmation has been recommended [25].

In 2008, *R. typhi* caused an outbreak in Texas, in which 73% of the confirmed patients were hospitalized and 27% were admitted to intensive-care units. This severity was explained by the delays in appropriate murine typhus treatment. Subsequently, the rate of hospitalization declined to 54% in 2009 after enhancement of notices about manifestations and appropriate treatments at the early stage of the disease from health authorities and the state medical society [26]. In other words, the disease burden of murine typhus might be reducible through early warnings or reminders. Previous studies have suggested the positive correlation between animal and human infections by murine typhus [26,27]. Perhaps the contemporaneous infectious situation in animal populations is useful information to develop an early warning system for zoonotic disease. Therefore, the objectives of this study were to use surveillance data from both small mammals and the human populace regarding *R. typhi* infection to clarify the relationship between the occurrence of indigenous murine typhus and the seropositive rate in animal hosts.

## Materials and methods

### Ethical statement

Human case records were retrieved from the Taiwan National Infectious Disease Statistics System administered by the Taiwan Centers for Disease Control (Taiwan CDC), and no personally identifiable information was used as part of this study. The animal surveillance of vector-borne zoonosis in international ports was implemented with the permission of the Taiwan CDC and complied with No. 33 Regulations Governing Quarantine at Ports in Taiwan (aligned with Article 20 and Annex 1 of the WHO International Health Regulations 2005). All animal trapping and handling procedures met Taiwanese legal requirements. The specimen collections from participants (suspected cases and vector animals of notifiable diseases) were authorized by Article 47 of Taiwan Communicable Disease Control Act and were exempt from Taiwan CDC institutional review board approval.

### Human surveillance system for murine typhus in Taiwan

Murine typhus has been designated a notifiable infectious disease since 2007. Physicians are required to notify health authorities of suspected murine typhus patients within 1 week of diagnosis based on clinical assessment with symptoms including headache, chills, fatigue, fever, body aches and rash. Blood and serum of the reported patients were collected for laboratory diagnosis by using a real-time polymerase chain reaction (PCR) test, the detection of *R. typhi*-reactive antibodies with indirect immunofluorescence assay (IFA) and organism isolation. Real-time PCR targeted the 17-kDa antigen-encoding gene in *Rickettsia* spp. The PCR products were sequenced and then assessed with the Basic Local Alignment Search Tool (https://www.ncbi.nlm.nih.gov) for resemblance to known species. For the IFA, each serum sample was applied to commercial (Focus Technologies, Inc., Cypress, CA, U.S.A.) or in-house slides coated with *R. typhi* antigens. A confirmed case was determined based on meeting one of the following criteria: (1) positive PCR test; (2) fourfold increase in *R. typhi*-specific immunoglobulin M (IgM) or IgG antibody levels in paired sera (each for the acute and

convalescent phase, with an interval >14 days); (3) positive for IgM at a 1:80 dilution and IgG at a 1:320 dilution; and (4) isolation of *R. typhi* [28].

## Surveillance on small mammals for *R. typhi* infection

Surveillance of zoonotic diseases, including *R. typhi* infection of small mammals, was launched at 25 international airports and harbors (Fig 1). Before 2014, trapping was undertaken each month at each site. Afterward, the frequency of trapping was changed to seasonally in March, June, September, and November (if positive, it will be monitored continuously for the following month). Each round of trapping lasted for one night. Sera were collected from trapped small mammals (rodents and shrews) and stored at −20°C for rickettsial infection tests. Each

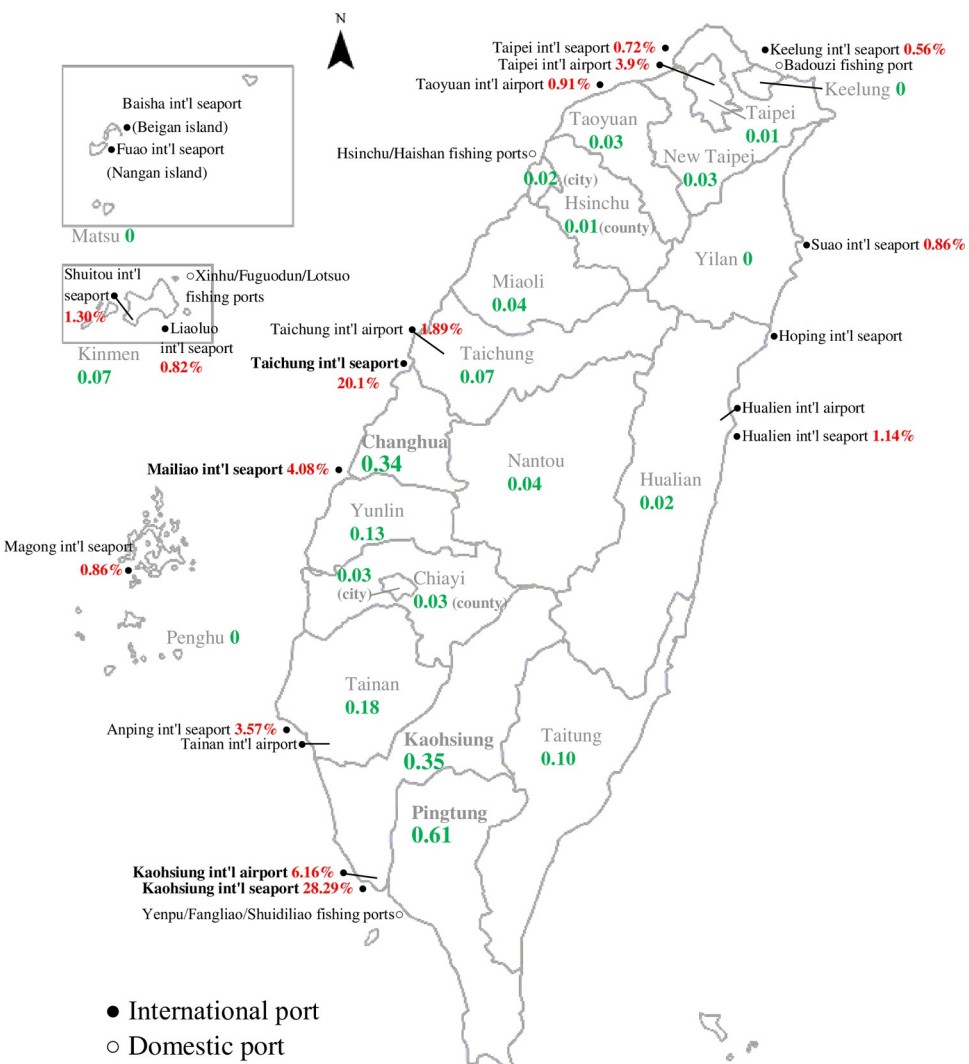

**Fig 1. Administrative district and study sites for the surveillance of *R. typhi* infections in small mammals and indigenous cases of murine typhus infection in humans in Taiwan.** The name of each administrative district is labeled in gray. International and domestic ports are labeled with solid and open circles, respectively. The seropositive rate of *R. typhi* in small mammals and the incidence rate (per 100,000 person-years) of indigenous murine typhus are marked with red and green, respectively. The base layer of the map was downloaded from National Land Surveying and Mapping Center of the Taiwan Ministry of the Interior, (https://maps.nlsc.gov.tw/MbIndex_qryPage.action?fun=8#).

serum sample was diluted 1:40 in phosphate-buffered saline (PBS) and applied to slides coated with *R. typhi* antigens (Focus Technologies, Inc., Cypress, CA, U.S.A.). PBS was used as a negative control. A serum sample positive for *R. typhi* served as a positive control. Diluted sera were added on IFA slides. The slides were then placed in a humid chamber at 37˚C for 30 min, soaked in PBS for 5 min, washed in deionized water, allowed to dry and covered with fluorescein isothiocyanate-conjugated goat anti-mouse IgG + IgA + IgM (H + L) (Zymed Laboratories, Inc., San Francisco, CA, U.S.A.) diluted 1: 40 in PBS. The slides were incubated again in a humid chamber at 37˚C for 30 min, washed in PBS, allowed to dry and fitted with coverslips. Assays were considered positive when the bright green fluorescence matched that of positive controls under a fluorescence microscope (Leica DM2500 fluorescence microscope; Leica Microsystems GmbH, Wetzlar, Germany).

## Statistical analysis

The incidence rates were calculated by dividing the number of confirmed cases of indigenous murine typhus (numerator) by the population in a period of time (denominator). The incidence rate was expressed as "cases per 100,000 person-years." Information on murine typhus-confirmed cases was downloaded from the Taiwan CDC website (http://www.cdc.gov.tw). Population size was represented by the result of the population calculation in each midyear (downloaded from the Department of Statistics of the Taiwan Ministry of the Interior, https://www.moi.gov.tw/stat/). In addition, the seropositive rate of *R. typhi* in small mammals was calculated based on the numbers of seropositive and tested samples. The seropositive rate of *R. typhi* in small mammals, number of cases and incidence rate of indigenous murine typhus for each month/year/region/rodent species (animal data) are shown in the S1–S3 Tables. The statistical association between the incidence of indigenous murine typhus in humans and the seropositive rate in small mammals was tested by Pearson's correlation test. Other statistical associations between two nominal variables were tested by $\chi^2$ test, Student's t test or Welch's t test, as appropriate. All statistical analyses were performed using Microsoft Excel (Microsoft, Washington, USA) and SPSS 22.0 (IBM, Illinois, USA).

## Results

### Surveillance of murine typhus

During 2007–2019, indigenous murine typhus human cases were continually detected in each of the 13 years (Fig 2). There were 380 total indigenous cases (average (±SD) of 29.23 (±8.76) cases/year) confirmed as domestic murine typhus, with an incidence rate (±SD) of 0.13 (±2.03*10⁻⁴)/100,000 person-years. During the study period, the average (±SD) of the mid-year population was 23,305,389 (±212,863.5) people in Taiwan per year. The annual number of indigenous infections and incidence rate ranged from 13~44 cases and 0.06~0.19/100,000 person-years, respectively. Across the study, 2007 and 2016 were the years with the most and fewest cases of indigenous murine typhus in Taiwan, respectively (Fig 3). The monthly distribution of indigenous murine typhus during 2007–2014 is shown in Fig 4. Overall, May, June and July were the months with the most cases; the number of patients with murine typhus detected in those three months accounted for 43.55% (108/248) of indigenous infections (accounted for 45.79%; 174/380, during 2007–2019). Moreover, the incidence rates in both May and June were higher than those in the other months (p<0.05). In contrast to the peak season during late spring/early summer, December and February were the months with the fewest number of indigenous cases, accounting for only 7.26% (18/248) of domestic infections (accounted for 7.11%; 27/380, during 2007–2019) (Fig 4). The incidence rates of those two months were both significantly lower than those of the other months (p<0.05).

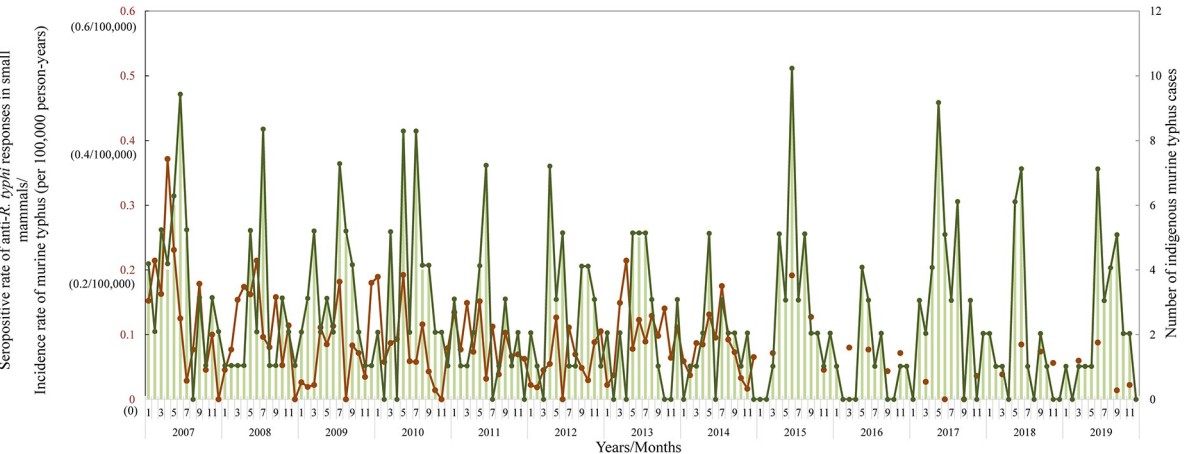

**Fig 2. Dynamics of *R. typhi* infection in small mammals and human cases of indigenous murine typhus from 2007–2019.** The seropositive rate of anti-*R. typhi* responses in small mammals and the incidence rate (per 100,000 person-years) of indigenous murine typhus cases are marked with brown and green, respectively. The numbers of indigenous murine typhus cases are shown as green bars.

Fig 1 shows that most confirmed human cases were reported in Pingtung (17.89%, 68/380), followed by Changhua (15.26%, 58/380) and Kaohsiung (33.42%, 27/380). The incidence rates per 100,000 persons of indigenous murine typhus in the three areas were 0.61, 0.34, and 0.35, respectively. In contrast, no indigenous cases were detected in Keelung, Yilan, or the Penghu and Matsu archipelagoes during the 2007–2019 surveillance.

## Surveillance of *R. typhi* infection in small mammals

A total of 6,796 animal sera were collected: 46.85% (3,184/6,796) were from *Rattus norvegicus*, 34.31% (2,332/6,796) from *Suncus murinus*, 10.58% (719/6,796) from *Rattus losea*, and 8.25% (561/6,796) from other small mammals. The seropositive rates of *R. norvegicus*, *S. murinus*, *R. losea* and others were 16.14% (514/3,184), 0.39% (9/2,332), 1.53% (11/719) and 4.63% (26/561), respectively. Overall, 560 sera exhibited an anti-*R. typhi* response, with a seropositive rate of 8.24%±0.33% (mean±SD).

2007 was the year with the highest seropositive rate of anti-*R. typhi* responses (14.26%; 69/484) in small mammals. In contrast, 2017 was the year showing the lowest seropositive rate (1.74%; 5/287) (Fig 3). In addition, according to the serological data from 2007–2014, when surveillance on small mammals was undertaken in each month, May (late spring in Taiwan) was the month with the highest seropositive rate of anti-*R. typhi* responses, which was 14.06% (72/512). The seropositive rate in May was significantly higher than those in other months (p<0.05). In contrast, November (early winter in Taiwan) was the month showing the lowest seropositive rate (5.19%; 23/443) over the same eight years, but statistical significance of the difference between seropositive rates in November and those in other months was not observed (Fig 4).

Graphically, the ports detected with the highest seropositive rates were Kaohsiung international seaport (28.29%; 408/1,442), Taichung international seaport (20.18%; 82/408), Kaohsiung international airport (6.16%; 17/276) and Mailiao international seaport (4.08%; 13/319) (Fig 1). The seropositive rates of the four sites were all higher than those of other ports (p<0.05). It is worth noting that the four ports with the highest seropositivity are all inside/near the areas with the highest incidence rate of indigenous murine typhus. The Kaohsiung international seaport and Kaohsiung international airport are located at the border region between Kaohsiung and Pingtung. The Taichung international seaport and Mailiao

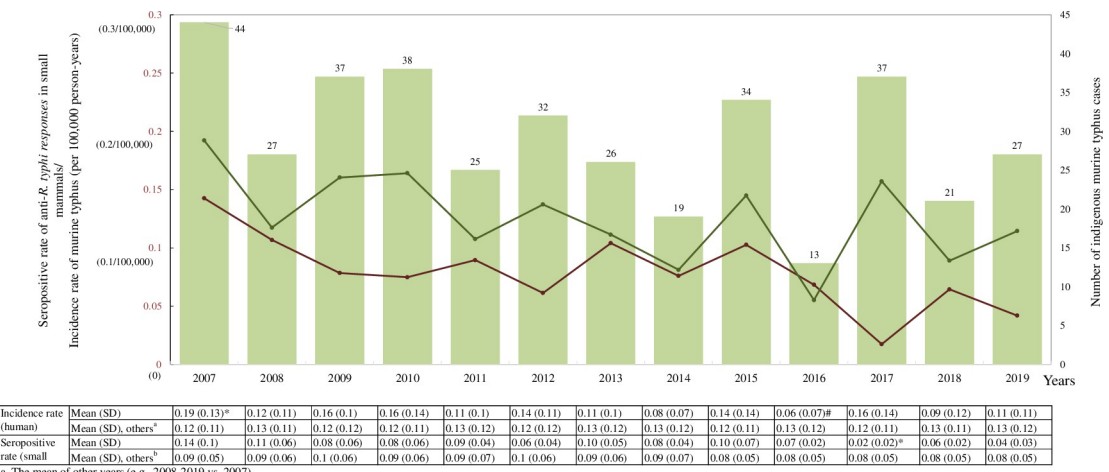

| Incidence rate (human) | Mean (SD) | 0.19 (0.13)* | 0.12 (0.11) | 0.16 (0.1) | 0.16 (0.14) | 0.11 (0.1) | 0.14 (0.11) | 0.11 (0.1) | 0.08 (0.07) | 0.14 (0.14) | 0.06 (0.07)# | 0.16 (0.14) | 0.09 (0.12) | 0.11 (0.11) |
|---|---|---|---|---|---|---|---|---|---|---|---|---|---|---|
| | Mean (SD), others[a] | 0.12 (0.11) | 0.13 (0.11) | 0.12 (0.12) | 0.12 (0.11) | 0.13 (0.12) | 0.12 (0.12) | 0.13 (0.12) | 0.13 (0.12) | 0.12 (0.11) | 0.13 (0.12) | 0.12 (0.11) | 0.13 (0.11) | 0.13 (0.12) |
| Seropositive rate (small | Mean (SD) | 0.14 (0.1) | 0.11 (0.06) | 0.08 (0.06) | 0.08 (0.06) | 0.09 (0.04) | 0.06 (0.04) | 0.10 (0.05) | 0.08 (0.04) | 0.10 (0.07) | 0.07 (0.02) | 0.02 (0.02)* | 0.06 (0.02) | 0.04 (0.03) |
| rate (small | Mean (SD), others[b] | 0.09 (0.05) | 0.09 (0.06) | 0.1 (0.06) | 0.09 (0.06) | 0.09 (0.07) | 0.1 (0.06) | 0.09 (0.06) | 0.09 (0.07) | 0.08 (0.05) | 0.08 (0.05) | 0.08 (0.05) | 0.06 (0.05) | 0.08 (0.05) |

a. The mean of other years (e.g., 2008-2019 vs. 2007).
b. Group 1 (2007-2014): The mean of other years when surveillance on small mammals was undertaken in each month (e.g., 2008-2014 vs. 2007). Group 2 (2015-2019): The mean of other years when surveillance on small mammals was undertaken seasonally in March, June, September, and November. (e.g., 2016-2019 vs. 2015).
*p<0.05 (Student's t-test)
#p<0.05 (Welch's t-test when unequal variances between compared groups)

**Fig 3. Yearly seropositive rate of anti-*R. typhi* responses in small mammals and incidence rate of indigenous murine typhus in humans from 2007–2019.** The seropositive rate and incidence rate (per 100,000 person-years) are shown as brown and green curves, respectively. The numbers of indigenous murine typhus cases are shown as green bars. The correlation between the incidence rate of indigenous human cases/seropositive rates in small mammals and years is presented in the table below.

international seaport are located near the northern and southern borders of Changhua, respectively (Fig 1).

## Correlation between the incidence rate in humans and the seropositive rate in small mammals

Ninety-six paired monthly rates (incidence rate of indigenous human murine typhus and seropositive rate of anti-*R. typhi* responses in small mammals) for the period from 2007–2014

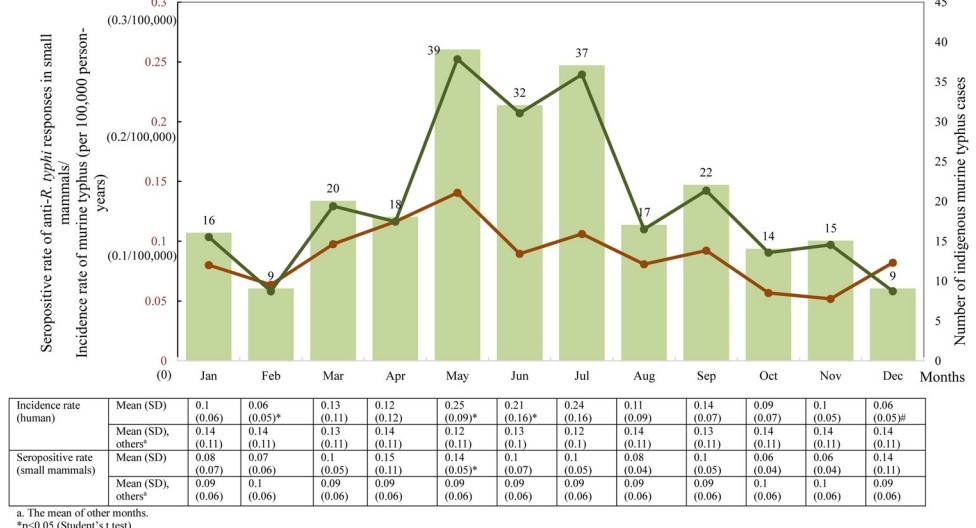

| Incidence rate (human) | Mean (SD) | 0.1 (0.06) | 0.06 (0.05)* | 0.13 (0.11) | 0.12 (0.12) | 0.25 (0.09)* | 0.21 (0.16)* | 0.24 (0.16) | 0.11 (0.09) | 0.14 (0.07) | 0.09 (0.07) | 0.1 (0.05) | 0.06 (0.05)# |
|---|---|---|---|---|---|---|---|---|---|---|---|---|---|
| | Mean (SD), others[a] | 0.14 (0.11) | 0.14 (0.11) | 0.13 (0.11) | 0.14 (0.11) | 0.12 (0.11) | 0.13 (0.1) | 0.12 (0.1) | 0.14 (0.11) | 0.13 (0.11) | 0.14 (0.11) | 0.14 (0.11) | 0.14 (0.11) |
| Seropositive rate (small mammals) | Mean (SD) | 0.08 (0.07) | 0.07 (0.06) | 0.1 (0.05) | 0.15 (0.11) | 0.14 (0.05)* | 0.1 (0.07) | 0.1 (0.05) | 0.08 (0.04) | 0.1 (0.05) | 0.06 (0.04) | 0.06 (0.04) | 0.14 (0.11) |
| | Mean (SD), others[a] | 0.09 (0.06) | 0.1 (0.06) | 0.09 (0.06) | 0.09 (0.06) | 0.09 (0.06) | 0.09 (0.06) | 0.09 (0.06) | 0.09 (0.06) | 0.09 (0.06) | 0.1 (0.06) | 0.1 (0.06) | 0.09 (0.06) |

a. The mean of other months.
*p<0.05 (Student's t test)
#p<0.05 (Welch's t test)

**Fig 4. Monthly seropositive rate of anti-*R. typhi* responses in small mammals and incidence rate of indigenous murine typhus in humans from 2007–2014.** The seropositive rate and incidence rate (per 100,000 person-years) are shown as brown and green curves, respectively. The numbers of indigenous murine typhus cases are shown as green bars. The correlation between the incidence rate of indigenous human cases/seropositive rates in small mammals and months is presented in the table below.

**Table 1. Correlation between the incidence rate of indigenous murine typhus in humans and seropositive rate of anti-*R. typhi* responses in small mammals from 2007–2014.**

| Types of paired incidence and seropositive rates [a] | Paired observation (n) | Incidence rate of murine typhus in humans [b] | Seropositive rate of anti-*R. typhi* responses in small mammals | R [c] | p value |
|---|---|---|---|---|---|
| synchronous | 96 | 0.13 (248/185,404,610*$10^6$)/ during 2007 Jan–2014 Dec | 8.89% (480/5,397*100%)/ during 2007 Jan–2014 Dec | 0.14 | 0.17 |
| 1-month delay | 95 | 0.13 (244/183,496,119*$10^6$)/ during 2007 Feb–2014 Dec | 8.91% (477/5,351*100%)/ during 2007 Jan–2014 Nov | 0.31 | **<0.01** |
| 2-month delay | 94 | 0.13 (242/181,587,627*$10^6$)/ during 2007 Mar–2014 Dec | 9.00% (476/5,290*100%)/ during 2007 Jan–2014 Oct | 0.37 | **<0.01** |
| 3-month delay | 93 | 0.13 (237/179,679,136*$10^6$)/ during 2007 Apr–2014 Dec | 9.06% (474/5,229*100%)/ during 2007 Jan–2014 Sep | 0.14 | 0.18 |

a. Synchronous or delayed (1~3 months behind the timing of the seropositive rate in small mammals) incidence rate of indigenous human cases of murine typhus were chosen for pairwise comparison.

b. Per 100,000 person-years

c. Pearson's correlation coefficient

(period of monthly surveillance on small mammals) were included in the calculation of Pearson's correlation coefficient for analyzing the relationship between infections in humans and small mammals. Although a synchronous correlation was not observed, the 1-month-delayed and 2-month-delayed incidence rates were both moderately positive correlated with the prior seropositive rate in small mammals. The 1-month-delayed and 2-month-delayed correlation coefficients were 0.31 (p<0.01) and 0.37 (p<0.01), respectively. In addition, the correlation coefficient decreased back to a nonsignificant level when the time of delay was further extended for 3 months (Table 1).

## Regression

To understand the risk of indigenous infection, regression models were further employed with domestic murine typhus-correlated factors that were detected in the univariate test or correlation analysis. After excluding the two factors with a negative association (December and February, the indigenous case numbers for the two months were both significantly fewer than those for others), the regression model with the remaining three positively correlated factors (May, June, and the seropositive rate in small mammals) could account for 19.36% and 25.10% of the incidence rate of indigenous human murine typhus in the following one and two months, respectively. Furthermore, if the area was limited to places with high incidence rates of indigenous infection (all the mentioned four ports with the highest seropositivity of anti-*R. typhi* responses are also inside/near these places), the regression model, which was constructed by the same variables, could still account for 16.16% and 18.23% of the incidence rate of indigenous human murine typhus in the subsequent one and two months, respectively (Table 2).

## Discussion

Murine typhus is distributed widely throughout the world. In Taiwan, documented detection of the disease can be traced back to the early 1990s [29,30]. Since 2007, when murine typhus was formally included as a notifiable disease in Taiwan, indigenous patients with this disease have been detected annually. Undoubtfully, murine typhus is an endemic disease in Taiwan. Regarding other countries, the incidence rates per 100,000 persons in Cyprus, Croatia, South Korea, Israel, and Spain are 24.5, 0.57, 0.08, 0.04, and 0.01, respectively. Interestingly, the Canary Islands of Spain have been reported to have a higher case number of murine typhus

**Table 2. Performance of different regression models for explaining the incidence rate of indigenous human cases of murine typhus.**

| Area | Port | Model | Included variables | N | R [c] | R² (%) | p value |
|------|------|-------|-------------------|---|-------|--------|---------|
| Nationwide | Each port in this study | Model 1 | SPR-1[a], May, Jun | 95 | 0.44 | 19.36 | <0.01 |
| | | Model 2 | SPR-2[b], May, Jun | 94 | 0.50 | 25.10 | <0.01 |
| Area with the highest incidence rate of indigenous human cases of murine typhus (Changhua, Kaohsiung, and Pingtung) | Four nearby ports (Taichung int'l seaport, Mailiao int'l seaport, Kaohsiung int'l seaport, and Kaohsiung int'l airport) | Model 1 | SPR-1[a], May, Jun | 95 | 0.40 | 16.16 | <0.01 |
| | | Model 2 | SPR-2[b], May, Jun | 94 | 0.43 | 18.23 | <0.01 |

a. SPR-1: seropositive rate in small mammals 1 month ahead of the timing of the incidence rate of indigenous human cases of murine typhus

b. SPR-2: seropositive rate in small mammals 2 months ahead of the timing of the incidence rate of indigenous human cases of murine typhus

c. Pearson's correlation coefficient

than other areas of Spain, Israel, or the United States [31–38]. The incidence rate of murine typhus was 0.13/100,000 persons in Taiwan during 2007–2019, which might reflect that the disease burden or the risk of outbreak is limited. In rats, the seropositive rates of anti-*R. typhi* responses in Canada, South Korea, Thailand, central area of Spain, Cyprus, and this study were 0.36%, 0.34%, 5.00%, 21.10%, 48.59% [39–43], and 8.24%, respectively. Due to the lack of identical antigens to define appropriate cutoff values, the serological results among these studies have not been compared and contrasted. Nevertheless, we observed that the seropositive rate of small rodents corresponded to human murine typhus cases in Taiwan during the 2007–2019 surveillance period.

In this study, most (46.85%) trapped small mammals in harbors and airports were identified as *R. norvegicus*. Similarly, the predominance of *R. norvegicus* has been addressed in other seaports of Indonesia [44], and a community of South Korea [45]. Compared to other small mammals in this study, *R. norvegicus* further demonstrated the highest seropositive rate of anti-*R. typhi* response. Although many *S. murinus* (34.31%) were also trapped in this study, the seropositive rate of *S. murinus* was extremely low. The difference might be attributed to the secondary antibodies used with different levels of reactivity between insectivore and rodent immunoglobulins. However, the difference between the seropositive rates could also be explained by diet, living habit and ectoparasitic infestation of *Suncus* species being distinct from those of *Rattus* species [46,47]. A review article summarized that fleas likely emerge with mammals and speciate with rodents, which have the most speciose extant fauna (74%). In contrast, only 8% of fleas are known to associate with insectivores, including *Suncus* species [46]. The principal vector of *R. typhi*, *Xenopsylla cheopis*, has also been observed to be ectoparasitic on *Rattus* rather than *Suncus* species [48]. Similar to this study, previous studies in Cyprus and Thailand have shown that *R. norvegicus* has a higher seropositive rate than *S. murinus* for *R. typhi* [41,43]. In addition, *Rattus* species having the highest seroprevalence of anti-*R. typhi* response was also observed in Egypt and South Korea [45,49]. These consistent observations imply that *R. norvegicus* seems to play an important role in the cultivation and transmission of *R. typhi*. Overall, we believe that *R. norvegicus* is an important animal reservoir that sustains the transmission of *R. typhi* with competent vectors in Taiwanese ports.

Seaports have been stated as the primary focus of murine typhus transmission [1]. In this study, we observed that most patients were located in Kaohsiung, Pingtung, and Changhwa. These areas are near the Kaohsiung seaport or Taichung seaport, which also revealed the highest seropositive rate in small mammals. Interestingly, the two seaports had the largest trading volume in Taiwan (Kaohsiung seaport and Taichung seaport dealt with 1,876,836,717 and

929,730,031 tons of international cargo, accounting for 54% and 27% of the traded volume in Taiwan, respectively) during 2007–2019 (Taiwan International Ports Corporation; http://www.twport.com.tw/). The high volume of traded cargo might provide an increased opportunity for vector and pathogen exchange from port to port. An Indonesian study demonstrated that people and rodents living in concordant urban areas showed a higher seropositive rate than the rate in populations from rural areas [50]. A close geographic association between seropositive animals and murine typhus patients has also been documented in Texas [26,51]. The geographic consistency or association between indigenous case onset and seropositivity in small mammals further suggests that the risk of infections could be similarly projected by both human and rodent hosts.

Seasonal variation in murine typhus incidence in tropical areas is not obvious, but in temperate regions, most detection of the disease is in hot and dry periods, such as late summer and early fall in California or late spring and early summer in Texas [8]. Taiwan is located in a subtropical region. Therefore, it was reasonable that we observed that most cases were detected in May, June and July (late spring/early summer) and the other months still had a few recorded cases. We suggest that murine typhus is a vector-borne zoonotic disease with intermediate seasonality in Taiwan. The intermediate degree was also supported by the results of surveillance in small mammals. In addition, the incidence of murine typhus associated with the abundance of fleas has been addressed [1,52]. Fleas are active and common in hot and warm seasons, and outdoor activity is also popular during this period. Therefore, the fact that most cases were detected in summer in this study might be explained by climate factors.

Infections in humans and rats are both vectored by fleas and correlated with the local density of this vector. Therefore, the correlation between human infections and seropositivity in rats is not surprising. In this study, human and animal infection rates indeed changed with similar dynamic trends. However, an interesting observation is that the correlation was accompanied by a time lag. We believe the delayed correlation observed in this study was not caused by chance, as the peak seasons of animal seropositivity and human infection both occurred during late spring/early summer. Furthermore, both animal seropositivity and human infection geographically focused in southern and central Taiwan. The life cycle of fleas includes different stages: eggs, larvae, pupae (covered with cocoons), and adults. Once the flea emerges from the cocoon, it immediately seeks a host to find a blood meal. Humans are accidental hosts of fleas, and wearing shoes, pants and clothes can further decrease the possibility of being bitten by fleas in nature. In contrast to humans, rodents frequently contact fleas, and common contact leads rodents to suffer from a higher risk of *R. typhi* infection in the environment. Therefore, the risk of *R. typhi* infection should be much more easily reflected in the small mammal population with frequent flea contact than in the human population. Due to the time lag of the delayed correlation observed from 1 to 2 months, surveillance in small mammals might provide an opportunity for the development of a reminder system of sporadic murine typhus in advance. Given the moderately positive correlation and limited number of confirmed domestic patients observed in this study, the actual effect of the proposed early reminder system is still needed to clarify with further studies.

Because rickettsiae or its nucleotides can be directly detected in blood in infected hosts for only a few [53], this limitation means that the molecular approach (e.g., PCR) likely will miss the brief duration of detection, making it difficult to convey the entire risk of *R. typhi* infection in cross-sectional surveillance with a limited sample size. Alternatively, detection of antibodies whose production was induced by previous infections becomes a substitute strategy in surveillance for risk assessment. Indeed, we observed several rises in seropositivity in small mammals accompanied by or followed by an ascending incidence rate of domestic murine typhus in this study.

For early detection of the possible risk of epidemics and timely control, early warning-oriented surveillance systems have been widely investigated and developed for infectious diseases. However, early warning-oriented surveillance systems (e.g., syndromic surveillance [54,55]) are mostly applied for early alerting of communicable diseases with high transmissibility, which can spread to a large number of individuals in a short time. In contrast, murine typhus cannot be effectively transmitted from humans to humans. The disease also cannot be transmitted widely by flying vectors such as mosquitos. Limited transmissibility accompanied by low incidence rates and the lack of specific symptoms of murine typhus all indicate that the development of a syndrome-based surveillance system would be difficult and inappropriate. Among the 380 indigenous cases detected in this study, 2 cases (0.53%) were fatal, and 40 cases (10.53%) were accompanied by severe manifestations, including pneumonia, acute hepatitis, renal insufficiency, meningitis, encephalitis or meningoencephalitis. Although reducing the limited occurrence of indigenous murine typhus might be challenging, lessening the severity of the disease might still be worth further effort. As early treatment with appropriate antibiotics for murine typhus could effectively shorten the duration of illness and reduce the risk of hospitalization and fatality, immediate or early alarm of rising risk might provide helpful information for public health on this easily neglected disease in clinics.

To date, there is no licensed preventive vaccine available for murine typhus worldwide. Environmental hygiene and appropriate protection from fleabites are effective ways to prevent infection. In Taiwan, harbors or airports where seropositivity is detected in small mammals are asked to implement mandatory enhanced environmental cleaning. However, *R. typhi* has spread worldwide and can be maintained for a long time in the classic rat–flea cycle [9]. Therefore, it is difficult to entirely eradicate the pathogen from the environment, especially in seaports with abundant trade activities. The sporadic occurrence of murine typhus is not surprising. The correlation of the occurrence of zoonotic disease between human and animal hosts has also been addressed in leptospirosis [56]. Interestingly, the delayed correlation observed in this study further provides an alternative strategy to develop a suitable alarm system through serosurveillance in small mammals. Based on this study, to shorten the duration of illness and reduce the risk of hospitalization and death caused by murine typhus in Taiwan, patients' exposure experience should be considered in clinics during May, June and the months after rising seropositivity in small mammals.

*R. typhi* serologically cross-reacts with other typhus group rickettsiae, such as *R. prowazekii* (the causative pathogen of epidemic typhus) [57]. Nevertheless, the lack of confirmed notification of epidemic typhus or isolation of *R. prowazekii* in Taiwan might also limit the bias in this study. However, the mild serological cross-reactivity between *R. typhi* and *R. felis* and the possible misdiagnosis [7,58,59] might cause nondifferential misclassification bias. Surveillance on animals in this study was based on the detection of anti-*R. typhi* responses but not the pathogen. Detected immunoglobulin types were not further differentiated. The proportion attributable to recent infections still needs to be further studied (e.g. using IgA and/or IgM-specific secondary antibodies).

While vector-borne pathogens are occasionally introduced into nonendemic regions, other factors are required for a pathogen to become established, such as the existence of local animal reservoirs, a susceptible human population, and competent vectors that can sustain the transmission of the pathogen where introduction occurred. However, humans are incidental and dead-end hosts for *R. typhi*. *R. typhi* needs other animal hosts for a sustainable presence. Hence, this suggests that *R. typhi* infection in rats occurs earlier than human infection in the same environment. In Taiwan, *R. norvegicus* is very common in cities and rural areas with high population densities. Murine typhus prevention by avoiding contact with rodents and/or

their excreta is highly suggested year round. In addition, murine typhus is worth considering slightly more in clinics during the period after increased seropositivity in small mammals.

## Supporting information

**S1 Table. Seropositive rate of anti-*R. typhi* responses in small mammals and incidence rate (per 100,000 person-years) of indigenous murine typhus for each month from 2007–2019.** (DOCX)

**S2 Table. Seropositive rate of anti-*R. typhi* responses in different small mammals for each study port from 2007–2019.** (DOCX)

**S3 Table. Incidence rate (per 100,000 person-years) of indigenous murine typhus for each administrative district from 2007–2019.** (DOCX)

## Acknowledgments

We appreciate Dr. Hsi-Chieh Wang for the establishment of the serosurveillance system in small mammals. We also thank the members of the Taipei, Northern, Central, Southern, Kaohsiung-Pingtung, and Eastern Regional Centers, Taiwan CDC, for their efforts in small mammal capture and collection of serum samples.

## Author Contributions

**Conceptualization:** Hwa-Jen Teng.

**Investigation:** Pai-Shan Chiang, Shin-Wei Su, Su-Lin Yang, Pei-Yun Shu, Wang-Ping Lee, Shu-Ying Li, Hwa-Jen Teng.

**Methodology:** Pei-Yun Shu, Wang-Ping Lee, Shu-Ying Li, Hwa-Jen Teng.

**Supervision:** Hwa-Jen Teng.

**Writing – original draft:** Pai-Shan Chiang.

**Writing – review & editing:** Hwa-Jen Teng.

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
