## [Decision Letter · Decision Letter 0]

14 Mar 2022

Dear Dr. Teng,

Thank you very much for submitting your manuscript "Delayed Correlation between the Incidence Rate of Indigenous Murine Typhus in Humans and the Seropositive Rate of Rickettsia typhi Infection in Small Mammals in Taiwan from 2007–2019" for consideration at PLOS Neglected Tropical Diseases. As with all papers reviewed by the journal, your manuscript was reviewed by members of the editorial board and by several independent reviewers. The reviewers appreciated the attention to an important topic. Based on the reviews, we are likely to accept this manuscript for publication, providing that you modify the manuscript according to the review recommendations. 

Dear Authors, 

All three reviewers have recommended to accept your submitted manuscript. Please read the comments by the reviewers below and make the minor modifications suggested by two of them.

Sincerely,

Manisha Biswal

Associate Editor

Hélène Carabin

Deputy Editor

Dear Authors, 

All three reviewers have recommended to accept the manuscript. Please read the comments by the reviewers below and make the minor modifications suggested by two of them.

Reviewer's Responses to Questions

**Key Review Criteria Required for Acceptance?**

**Methods**

-Are the objectives of the study clearly articulated with a clear testable hypothesis stated?

-Is the study design appropriate to address the stated objectives?

-Is the population clearly described and appropriate for the hypothesis being tested?

-Is the sample size sufficient to ensure adequate power to address the hypothesis being tested?

-Were correct statistical analysis used to support conclusions?

-Are there concerns about ethical or regulatory requirements being met?

Reviewer #1: Yes

Reviewer #2: (No Response)

Reviewer #3: Accept

**Results**

-Does the analysis presented match the analysis plan?

-Are the results clearly and completely presented?

-Are the figures (Tables, Images) of sufficient quality for clarity?

Reviewer #1: Poor figure resolution, difficult to review

Reviewer #2: (No Response)

Reviewer #3: Accept

**Conclusions**

-Are the conclusions supported by the data presented?

-Are the limitations of analysis clearly described?

-Do the authors discuss how these data can be helpful to advance our understanding of the topic under study?

-Is public health relevance addressed?

Reviewer #1: It is to be noted that this study demonstrate only a moderate co-relation with occurrence of human cases corresponding to sero-positive rate in small mammals at 1 or 2 month prior. Given the low sample size [human cases (n=380 over 13 years) as well as sero-positive small mammals (only 8%)], it is difficult to interpret the effect of this co-relation in actual scenario. It is therefore suggested to highlight limitations of the present study. The term ‘moderately positive co-relation’ should be used for emphasizing the results.

Reviewer #2: (No Response)

Reviewer #3: Accept

- I agree that there are limitations to the molecular approach.

If there is an analysis result of a gene directly detected for R. typhi in blood, flea, host, etc., it is thought that it will be possible to reveal evidence for the exchange of vectors and pathogens from port to port according to trading countries. If you can provide more results for the molecular approach, please provide additional information.

**Editorial and Data Presentation Modifications?**

Reviewer #1: (No Response)

Reviewer #2: (No Response)

Reviewer #3: Accept

**Summary and General Comments**

Reviewer #1: Comments:

1. It is to be noted that this study demonstrate only a moderate co-relation with occurrence of human cases corresponding to sero-positive rate in small mammals at 1 or 2 month prior. Given the low sample size [human cases (n=380 over 13 years) as well as sero-positive small mammals (only 8%)], it is difficult to interpret the effect of this co-relation in actual scenario. It is therefore suggested to highlight limitations of the present study. The term ‘moderately positive co-relation’ should be used for emphasizing the results. 

2. Surveyed region wise as well as year wise positivity data on human and small mammals (indicating the rodent species too) should be presented, preferably in a table format. 

3. Figure resolution needs to be increased.

Reviewer #2: Serosurveillance of murine typhus were conducted monthly in small mammals in airports and seaports throughout Taiwan during 2007 and 2014. When compared with monthly human case numbers in Taiwan during the same time period, a significant positive correlation has been observed. Interestingly, the correlation is one to two months delayed, making increase of seropositive rate in small mammals a potentially useful predictor to alert for human murine typhus infection. The data were clearly presented and the conclusions are well supported. Several minor suggestions to consider: 

1) The correlation is established using monthly data from the entire Taiwan. Given the large variations of human infection incidents and rodent seropositive rates across Taiwan, it might be worthwhile to further investigate sub-regions, especially those having geographical overlapping of sampling sites. 

2) Choice of secondary antibody for IFA assay of animal serum included mixture of IgG/IgA/IgM. Since IgG has a much longer half life, using specific IgA and/or IgM secondary antibodies might better differentiate more recent infections versus infections that occurred months ago.

Reviewer #3: (No Response)

PLOS authors have the option to publish the peer review history of their article (what does this mean?). If published, this will include your full peer review and any attached files.

Reviewer #1: Yes: Dr. Siraj Ahmed Khan

Reviewer #2: No

Reviewer #3: No

Figure Files:

Data Requirements:

Reproducibility:

References

---

## [Editor Report · Decision Letter 1]

5 Apr 2022

Dear Dr. Teng,

We are pleased to inform you that your manuscript 'Delayed Correlation between the Incidence Rate of Indigenous Murine Typhus in Humans and the Seropositive Rate of Rickettsia typhi Infection in Small Mammals in Taiwan from 2007–2019' has been provisionally accepted for publication in PLOS Neglected Tropical Diseases.

Best regards,

Manisha Biswal

Associate Editor

Hélène Carabin

Deputy Editor

The authors have addressed the minor revisions recommended by the reviewers.

---

## [Editor Report · Acceptance letter]

20 Apr 2022

Dear Dr. Teng,

We are delighted to inform you that your manuscript, "Delayed correlation between the incidence rate of indigenous murine typhus in humans and the seropositive rate of Rickettsia typhi infection in small mammals in Taiwan from 2007–2019," has been formally accepted for publication in PLOS Neglected Tropical Diseases.

Best regards,

Shaden Kamhawi

co-Editor-in-Chief

Paul Brindley

co-Editor-in-Chief
